# Association and Effectiveness of PAX1 Methylation and HPV Viral Load for the Detection of Cervical High-Grade Squamous Intraepithelial Lesion

**DOI:** 10.3390/pathogens12010063

**Published:** 2022-12-30

**Authors:** Mingzhu Li, Chao Zhao, Yun Zhao, Jingran Li, Xiaobo Zhang, Wei Zhang, Qingqing Gao, Lihui Wei

**Affiliations:** Department of Obstetrics and Gynecology, Peking University People’s Hospital, No. 11 Xizhimen South Street, Beijing 100044, China

**Keywords:** high-grade squamous intraepithelial lesion, PAX1, methylation, high risk human papillomavirus, viral load

## Abstract

Background: PAX1 methylation (PAX1^m^) and HPV viral load (VL) have been reported to detect cervical high-grade squamous intraepithelial lesions (HSIL), but the relationship between them is unclear. This study aimed to evaluate the correlation between HPV VL and PAX1^m^ and its effectiveness in predicting cervical lesions. (2) Methods: A total of 476 women referred to colposcopy for abnormal cervical screening at the Peking University People’s Hospital between November 2020 and November 2021 were enrolled. PAX1^m^ and HPV VL were determined by QMSP and BMRT-HPV reports type-specific VL/10,000 cells, respectively. (3) Results: PAX1^m^ was significantly increased in HSIL, especially in cervical cancer, but there was no significant difference between cervical intraepithelial neoplasms 1(CIN1) and CIN2. However, HPV VL significantly differed between CIN1 and CIN2 but not between CIN3 and cervical cancer. In general, PAX1^m^ positively correlated with all hrHPV VL, mainly in the HPV16/18 VL (*p* < 0.001), but had no relationship with the other 12 types of hrHPV VL. PAX1^m^ had the highest specificity in diagnosing CIN2+, followed by HPV16/18 VL, which are higher than cytology ≥ASCUS. (4) Conclusions: Hypermethylation of PAX1 is associated with high HPV VL, especially HPV16/18, and both present advantageous specificity in detecting CIN2+.

## 1. Introduction

Persistent infection with a high-risk type of human papillomavirus (hrHPV) can lead to cervical precancerous lesions and invasive cancer of the uterine cervix. Primary hrHPV testing is currently accepted for cervical screening over cytology based on its superior sensitivity and greater reassurance of negative predictive value for high-grade squamous intraepithelial lesions (HSIL) [1]. However, while most hrHPV infections are transient and harmless with productive infections and do not give rise to cervical cancer, it is less specific and has low positive predictive value compared to cytology [2]. Therefore, a triage test is required to distinguish progressing HPV infections from transient infections.

New biomarkers, such as methylation of host or HPV genes, immunohistochemical staining of cervical smears for tumor markers, such as p16^INK4a^ [3], and the use of HPV viral load (VL), are being evaluated as alternative triage methods. The relationship between hrHPV VL and the progression of cervical lesions remains controversial. Some studies have demonstrated an association between high VL and the severity of cervical lesions [4,5], whereas others have reported either the opposite result or no clear relationship [6,7]. Duan LF et al. reported that BioPerfectus Multiplex Real-Time PCR(BMRT) hrHPV VL increased with the grade of cervical lesions and is as sensitive as Cobas4800 for primary cervical cancer screening [8]. DNA methylation has been extensively studied for the early diagnosis of cancers and prognosis prediction in cancer patients. The paired box gene1 (PAX1) has been reported as a potential methylation biomarker for its promise in the detection of cervical intraepithelial neoplasms (CIN) grade 3 and worse lesions (CIN3+) with high sensitivity and specificity [9]. Moreover, incorporating the detection of PAX1 methylation (PAX1^m^) with an hrHPV test could improve the efficacy of cervical cancer screening [10]. However, the correlation between PAX1^m^ and HPV VL in the progress of cervical lesions has not been reported. Therefore, the objective of this study was to clarify the correlation between PAX1^m^ and HPV VL in patients with different degrees of cervical abnormality and to explore the clinical value of PAX1^m^ and HPV VL for the predictive diagnosis of cervical lesions.

## 2. Materials and Methods

### 2.1. Study Subjects

We selected 510 patients referred for colposcopy with abnormal cervical screening in Peking University People’s Hospital between November 2020 and November 2021. Cervical cells were collected before colposcopy for PAX1^m^ and BMRT hrHPV VL testing. Those with a previous history of hysterectomy or who refused to enroll in the study were excluded. Pathologic confirmation was performed by colposcopy-directed punch biopsy. The endpoint was CIN2+, which has been used in most studies to evaluate the parameter performance. After the exclusion of 21 women with incomplete data and 15 women histologically confirmed with adenocarcinoma in situ (AIS), the final data of 476 women were evaluated and analyzed. All participants signed an informed consent form. The study was approved by the Institutional Review Board of Peking University People’s Hospital (2020PHB298-01).

### 2.2. Quantitative Methylation-Specific PCR

Cervical exfoliated cells were centrifuged and stored in phosphate-buffered saline at −20 °C until testing. Genomic DNA was extracted using standard protocols and converted to bisulfite form using the EZ DNA Methylation-Gold kits (Zymo Research, Irvine, CA, USA). Methylation-specific PCR was performed on the Light Cycler LC480 system (Roche Applied Science, Penzberg, Germany) to determine the methylation level of PAX1 according to the manufacturer’s instructions (Hoomya Ltd., Changsha, China). The type II collagen gene (COL2A) was used as an internal reference. The (delta Cp(ΔCp)ΔCp is the difference between the ΔCp values for PAX1 and COL2A. DNA methylation status was calculated based on the differences between the Cp values of the tested and referred genes: ΔCp = Cp target gene—Cp Col2A. A low ΔCp value indicated a high PAX1^m^ level in the collected samples.

### 2.3. BMRT HPV PCR Assay

The BMRT is detected based on a PCR-based high-risk HPV assay performed with the fluorescence-based multiplex HPV DNA genotyping kit (Bioperfectus Ltd., Taizhou, China). PCR primers and corresponding TaqMan probes were developed for the 21 most prevalent HPV types to amplify the HPV L1 gene, including 14 hrHPV genotypes (HPV16,18,31,33,35,39,45,51,52,56,58,59,66,68), and 7 medium risk and low risk-HPV genotypes (HPV26,53,82,73,6,11,81). For this study, a 14-type high-risk BMRT assay was used. A single-copy gene encoding DNA topoisomerase III (human TOP3) was amplified in the reaction to control DNA quality and determine the relative viral copy numbers in the samples. The normalization of HPV type-specific VLs was performed as follows: VL = log10[CnHPV/Cn TOP3) × 10,000] copies/10,000 cells, where Cn HPV is the quantity of HPV DNA, and Cn TOP3 is the number of human cells. The experimental procedure was conducted according to the kit manufacturer’s instructions. The detailed process was described by Dong and Duan [8,11].

### 2.4. Statistics

Statistical analysis was performed using SPSS 26.0 (IBM Corp., Armonk, NY, USA). Categorical variables were presented as frequencies and percentages (Histological and pathological results, cervical cytology, hrHPV genotype, etc.). Quantitative data were described by mean ± standard deviation (age). Non-normally distributed continuous variables presented as median and Inter-quartile range (IQR), the indicators including all hrHPV VL[Log(all-hrHPV)], HPV16/18 VL[Log(HPV16/18)], other 12 types of hrHPV VL[Log(other-hrHPV)] and PAX1^m^. The Mann -Whitney U test was used to compare the continuous variables between two independent groups. The Spearman correlation analysis was used to test the correlation between VL and gene methylation indexes in different histopathological results. Continuous variables (PAX1^m^, all hrHPV VL, HPV16/18 VL, and the other 12 types of hrHPV VL) were stratified into four intervals to form grade variables and assigned Negative (0), Low (1), Moderate (2), and High (3). A logistic regression model for CIN2+ was constructed to calculate ORs and 95%CI. Then the ordered categories as a one degree-of-freedom were put into the regression model again for P–trend. Receiver operating characteristic (ROC) curves were used to assess the area under the curve (AUC) of PAX1^m^, all hrHPV VL, and HPV16/18 VL, differentiating between < CIN2/CIN2+. The optimal critical value to distinguish < CIN2/CIN2+ was determined according to the maximum principle of the Youden index (YI). Sensitivity (Se), specificity (Sp), positive predictive value (PPV), negative predictive value (NPV), odds ratio (OR). Clopper-Pearson confidence intervals were used for sensitivity and specificity. Confidence intervals for predictive values reported by Mercaldo et al. [12] were used. We tested differences in sensitivity and specificity using McNemar’s χ^2^ test, differences in PPV, NPV, and absolute risk with the method described by Leisenring W [13], and differences in OR with the method described by Altman DG [14]. Two-tailed *p*-values of less than 0.05 were considered statistically significant.

## 3. Results

### 3.1. Overview of the Data

The median age of the 476 women was 40 years (range 34–49 years). The BMRT assay revealed that 285 (59.9%) of the women had a single HPV infection, 106 (22.3%) had multiple infections, 193 (40.5%) of the women were HPV16/18+, and 198 (41.6%) had other 12 types of hrHPV type. As for the pathological results, there were 100 cases of chronic cervicitis, 57 cases of CIN1, 118 cases of CIN2, 171 cases of CIN3, and 30 cases of cervical cancer. Table 1 shows the distribution of CIN, cervical cytology results, hrHPV status, and HPV genotype.

### 3.2. Correlation between PAX1^m^, hrHPV Viral Load, and Cervical Lesions

Figure 1 showed that the PAX1^m^ levels were statistically different in different degrees of cervical lesions. PAX1^m^ was significantly increased in HSIL and above, especially in cervical cancer (*p* < 0.001), although there was no significant difference between CIN1 and CIN2(*p* = 0.334). HPV VL also increased with CIN progression, and there was a significant difference between CIN1 and CIN2 but not between CIN3 and cervical cancer. The HPV16/18 VL was significantly higher in women with CIN3 than in those with CIN2 (*p* < 0.05), but there was no significant increase in the other 12 types of hrHPV VL (Appendix A).

Regardless of the severity of the cervical lesion, PAX1^m^ ΔCp levels were constantly negatively correlated with all hrHPV and HPV16/18 VL (rs = −0.253 and −0.269; *p* < 0.001). It is suggested that PAX1 hypermethylation has a high VL of all hrHPV, and HPV16/18 in particular, but had no significant difference in the other 12 types of hrHPV VLs (*p* = 0.948) (Table 2).

### 3.3. Association of HPV Genotype, PAX1^m^, hrHPV Viral Load with CIN2+ Risk by Different Grade Variables

We observed a striking association for risk of CIN2+ with different HPV genotype, HPV16/18+ (OR: 8.89) > HPV31/33/52/58+ (OR: 5.25) > Other HPV+ (OR: 1.44). The OR for CIN2+ increased at Low [11.49 (10.45–16.26)], Moderate [9.21 (7.74–10.41)], and High [5.69 (1.09–7.72)] intervals of PAX1^m^, when the interval with 5.69 (1.09–7.72), OR for CIN3+ reach up to 15.74 (8.11–30.55). With regards to the VL of all hrHPV and HPV16/18, when the moderate intervals was [4.75 (4.10–5.29)] and [4.76 (4.06–5.25)], the OR of CIN2+ were 10.08 (5.25–19.35) and 7.54 (3.15–18.04), which was higher than that with low intervals [3.24 (1.83–5.73) and 2.16 (1.18–3.94)]. While in the high intervals with [6.02(5.31–8.53)] and 5.93 (5.26–8.21), the OR of CIN2+ decreased to 5.03 (2.79–9.07) and 4.29 (2.10–8.76), and the same trend with other 12 types of hrHPV VLs was observed (Table 3).

### 3.4. Performance and Cutoff Values of PAX1^m^ and HPV Viral Load Assay for the Incidence of CIN2+

According to the ROC curves, the optimal cutoff of PAX1^m^ for identifying CIN2+ was 10.63, with an AUC of 0.64 (95% CI = 0.0.59–0.69; *p* < 0.001), respectively. Regarding HPV VLs, all hrHPV VL had an optimal cutoff of 3.32 copies/10,000 cells (log10-transformed) for detecting CIN2+ (AUC of 0.67, 95% CI = 0.61–0.72; *p* < 0.001). HPV16/18 VLs had an AUC of 0.66 (95% CI = 0.60–0.71; *p* < 0.001) and an optimal cutoff of 2.34 copies/10,000 cells for identifying CIN2+, see Figure 2.

To identify CIN2+, the sensitivity of PAX1^m^ (cut off ≤ 10.63) (Se: 46.4%) is lower than cytology (ASCUS+) (Se: 54.5%), hrHPV test (Se: 90.6%), and hr-HPV VL (cut off > 3.32 copies per 10,000 cells) (Se: 78.1%), but comparable to the HPV16/18 genotype (Se: 50.2%) and HPV16/18 VL (cut off > 2.34 copies per 10,000 cells) (Se: 48%). However, PAX1^m^ (cut off ≤ 10.63) had the highest specificity (Sp: 86.6%) in diagnosing CIN2+, followed by HPV16/18 VL (cut off > 2.34 copies per 10,000 cells) (Sp: 84.1%), both of them higher than cytology ≥ASCUS(Sp: 72.6%) (Table 4).

## 4. Discussion

In our study, we found that hypermethylation of PAX1 is associated with a high viral load of hrHPV, especially HPV16/18. Women with hypermethylation of PAX1 had a significantly higher incidence of CIN3, especially cervical cancer, compared with CIN2 and below, but there was no significant difference between CIN1 and CIN2, indicating its potential role in the late status of cervical lesions. On the other hand, HPV VL had a significantly higher incidence of CIN2+ compared with CIN1 and below, but once processed to CIN3, HPV VL(except HPV16/18) did not increase with disease progression anymore, indicating that the replication and release of HPV particles was replaced by active proliferation of the disease at advanced stages of the cervical lesion. We also demonstrated that PAX1^m^ and HPV16/18 VL exhibited higher specificity than cytology in detecting CIN2+.

An association between hrHPV VL and the severity of cervical disease was first described in 1999 [15]. Since then, hrHPV VL has been introduced as a marker for persistent infection and progression to CIN, to distinguish between regressing CIN2 and CIN3 lesions [16] and to predict progression to cervical cancer [17]. In our study, low-grade lesions(≤CIN1) had lower HPV VLs than high-grade lesions(≥CIN2) by comparing all hrHPV VLs. When HPV VLs were stratified into four intervals, the OR of CIN2+ in moderate intervals was higher than in high intervals (10.08 vs. 5.03), indicating that a higher HPV VL does not mean a higher risk of cervical lesion.

Different HPV genotypes may affect the relationship between hrHPV VL and the severity of cervical lesions. A recent prospective cohort study in China found cumulative risk of CIN2+ had a close correlation with the HPV A9 group (HPV−16, −31, −33, −35, −52, −58) rather than the A7 group (HPV−18, −39, −45, −59, −68) [11]. Some studies reported that HPV16 VL could either differentiate between HSIL(CIN2+) and LSIL(CIN1) or between cervical cancer and lower grades of disease [18,19]. However, other reports concluded that HPV16 VL is not associated with the severity of the disease [20]. In our study, HPV VL also increased with the severity of cervical lesions, and especially there was a significant difference between low-grade and high-grade lesions (*p* < 0.001), but not between high-grade lesions and cervical cancer. We found that HPV16/18 VL was positively correlated with the grade of cervical lesions (CIN3 > CIN2 > CIN1, *p* < 0.05) but had no significant difference between CIN3 and cervical cancer. In addition, no significant correlation was found between the other 12 types of hrHPV VL and cervical lesions, but still of great significance was differentiating CIN2+ from CIN1 and below lesions, just as Wang M et al. reported [21]. It is also worth mentioning that when we delved into what plays a key role in the HPV16/18 portfolio, we supposed that HPV16 might play the decisive role. However, the number of HPV18 cases was too limited, comparing 23 vs. 173 with HPV16, so it was necessary to expand the sample further (Appendix A).

DNA methylation is another potential factor in the malignant transformation of the HPV-infected epithelium. PAX1 is a tumor suppressor gene, and high PAX1methylation levels induce tumorigenesis, such as cervical cancer [22]. PAX1^m^ has been extensively studied for its clinical application for cervical cancer screening [23]. As a molecular triage tool, PAX1^m^ showed comparable clinical performance to cytology and better specificity than the HPV16/18 genotype in detecting CIN2+, which is promising for early detection of CIN and predicting disease progression in hrHPV+ women [24]. In addition, PAX1^m^ can be used for screening women with atypical squamous cells of undetermined significance (ASCUS) and has also shown better diagnostic performance than HPV-DNA in predicting high-grade CIN (CIN2/3) [25]. PAX1^m^ has also been reported to predict the efficacy of concurrent chemo-radiotherapy in cervical cancer [26]. In this study, PAX1^m^ increased significantly in CIN3 and strikingly the highest in SCC, but could not distinguish CIN2 from CIN1. In addition, when PAX1^m^ was stratified into four intervals, the OR of CIN2+ increased with the increase of grade variables, further indicating that this highly methylated PAX1 was probably associated with the late status of the disease.

It has been reported that HPV VL could differentiate between normal and abnormal cytology with a sensitivity of 75% and a specificity of 80%. With some variation between different genotypes, methylation levels could differentiate normal and low-grade cytology from high-grade cytology with a sensitivity of 64% and a specificity of 82% [27]; however, the relationship between VL and methylation status was not clarified. In our study, a positive correlation between PAX1^m^ and hrHPV VL was observed, particularly with HPV16/18 VL, but had no significant difference in the other 12 types of hrHPV VLs. The present data suggest that high PAX1 methylation accompanied by high VL in the oncogenic pathway might be HPV16/18 genotype related.

In 2020, the American Cancer Society (ACS) proposed updated screening guidelines that recommended women initiating cervical cancer screening at the age of 25 years and undergoing primary HPV testing every 5 years until the age of 65 [28]. Despite its high sensitivity, hrHPV testing cannot distinguish whether or not an HPV-positive result is associated with a cervical precancerous lesion, and a positive hrHPV result may lead to over-interpretation of minor cellular abnormalities; therefore, a highly specific triage method is necessary. Cytology is considered appropriate for application as a reflex test for hrHPV-positive women due to its high specificity. In our study, for detecting CIN2+, the sensitivity of cytology alone was 54.5% (95% CI, 48.9–60.1), and the specificity was 72.6%, which was lower than PAX1^m^(cut off ≤ 10.63), HPV16/18 genotype and HPV16/18 VL(cut off > 2.34 copies per 10,000 cells) [86.6% (95% CI, 80.3–91.5) vs. 79.0% (95% CI, 71.8–85.1) vs. 84.1% (95% CI, 77.4–89.4)]. There was no statistically significant difference of specificity in the diagnosis of CIN2+ between PAX1^m^, HPV16/18 genotype, and HPV16/18 VL (*p* > 0.05), indicating both of them present advantageous specificity in detecting CIN2+.

This study has limitations. It was performed at a single center, and the sample size was relatively small, mainly because the participants were all from those referred to colposcopy with abnormal cervical screening. The correlation between PAX1 methylation and HPV VL and the relationship with the disease severity was not analyzed for each HPV genotype, as the number of cases involving each type of HPV would be smaller. In addition, HPV integration has not been explored. Some researchers believe that samples containing integrated HPV tend to have lower HPV VLs than that in purely episomal or mixed forms [19,29], although this is not always the case, especially with HPV18 and HPV58 [30,31]. Furthermore, viral integration had been found throughout the course of the disease but was poorly associated with cervical disease [27]. Nevertheless, understanding the relationship between HPV integration, HPV VL, and methylation is something we are interested in exploring in the future.

## 5. Conclusions

Hypermethylation of PAX1 is associated with high HPV VL, especially HPV16/18, and both of them present advantageous specificity in detecting CIN2+. HPV16/18 VL appears to have an overall specificity of 84.1% (95%CI: 77.4–89.4%) and PAX1 methylation with a specificity of 86.6% (95%CI: 80.3–91.5%) for detecting CIN2+.

## Figures and Tables

**Figure 1 pathogens-12-00063-f001:**
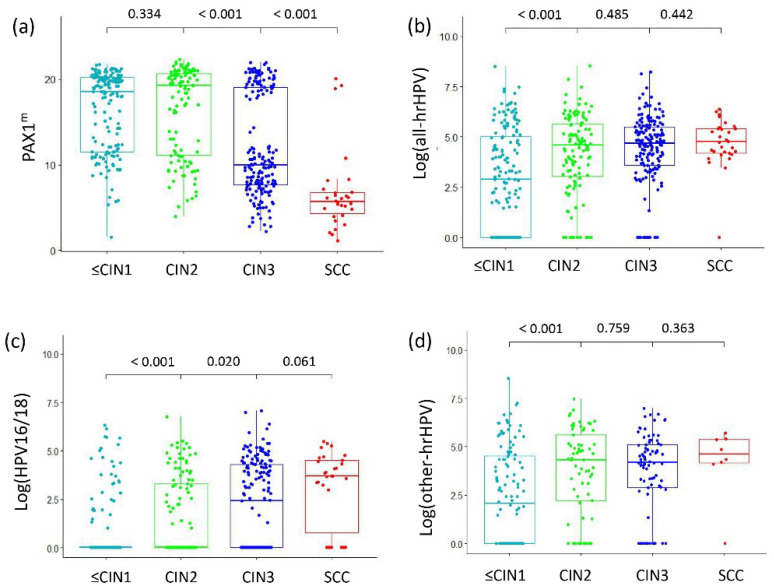
Distribution of PAX1^m^ and hrHPV VL in different degrees of cervical lesions. (**a**) PAX1^m^; (**b**) all hrHPV VL; (**c**) HPV16/18 VL; (**d**) other 12 types of hrHPV VL.

**Figure 2 pathogens-12-00063-f002:**
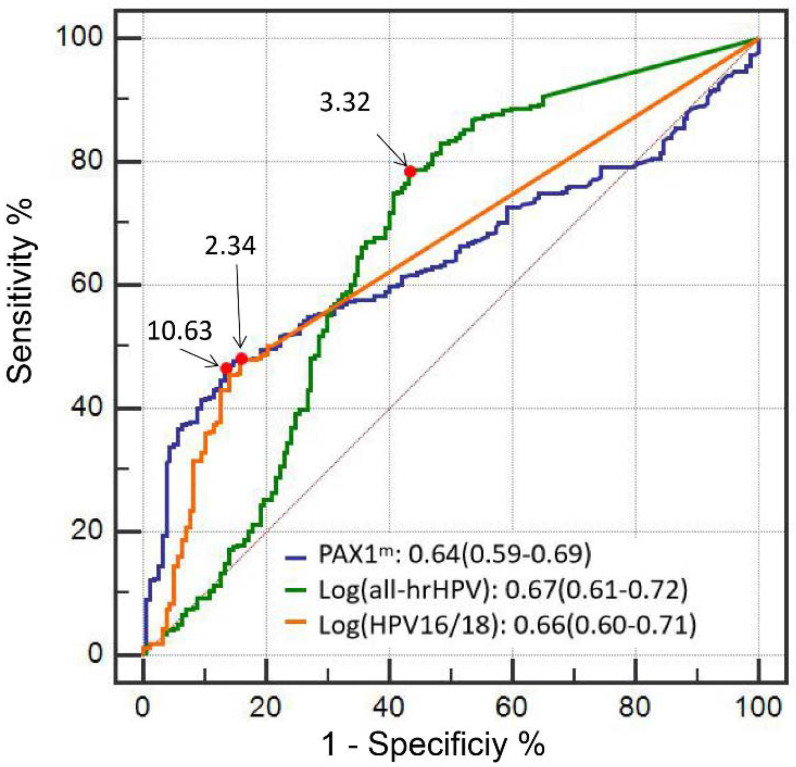
Cutoff value of PAX1^m^ and hrHPV viral load in detecting CIN2+.

**Table 1 pathogens-12-00063-t001:** Clinical characteristics.

	Frequency (*n*)	Proportion (%)
Biopsy		
-Normal	100	21.0
-CIN1	57	12.0
-CIN2	118	24.8
-CIN3	171	35.9
-SCC	30	6.3
Cytology		
-NILM	259	54.4
-ASCUS	62	13.0
-LSIL	35	7.4
-ASC-H	39	8.2
-HSIL	70	14.7
-AGC	9	1.9
-SCC	2	0.4
Multiple HPV types		
-yes	106	22.3
-no	285	59.9
-negative	85	17.9
hrHPV		
-HPV16/18(+)	193	40.5
-HPV31/33/52/58(+)	139	29.2
-other hrHPV type(+)	59	12.4

CIN: cervical intraepithelial neoplasia; NILM: negative for intraepithelial lesion or malignancy; ASCUS: atypical squamous cells of undetermined significance; LSIL: low-grade squamous intraepithelial lesion; ASC-H: atypical squamous cells cannot exclude HSIL; HSIL: high-grade squamous intraepithelial lesion; AGC: atypical glandular cell; SCC: squamous cell carcinoma.

**Table 2 pathogens-12-00063-t002:** Correlation between PAX1^m^ and hrHPV VL in different degrees of cervical lesions.

Viral Load	PAX1^m^							
	≤CIN1		CIN2		CIN3+		Total	
	rs *	*p*	rs	*p*	rs	*p*	rs	*p*
Log(all-hrHPV)	−0.142	0.076	−0.231	0.012	−0.270	<0.001	−0.253	<0.001
Log(HPV16/18)	−0.221	0.005	−0.211	0.022	−0.144	0.041	−0.269	<0.001
Log(other-hrHPV)	0.011	0.892	−0.084	0.365	−0.082	0.250	−0.003	0.948

* Spearman correlation coefficient.

**Table 3 pathogens-12-00063-t003:** Stratified intervals of HPV genotype, PAX1^m^, viral load of all hrHPV, HPV16/18, and other HPV genotypes at risk of CIN2+ (*n* = 476).

	OR(95%CI) ^1^				*p*-Trend ^2^
HPV Genotype	Negative *n* = 85	Other HPV(+) *n* = 59	HPV31/33/52/58(+) *n* = 139	HPV16/18(+) *n* = 193	
OR, CIN2+	1.0	1.44 (0.73–2.85)	5.25 (2.92–9.41)	8.89 (4.97–15.90)	<0.001
*p*		0.289	<0.001	<0.001	
PAX1^m^	Negative *n* = 230	Low *n* = 82	Moderate *n* = 82	High *n* = 82	
Median (range)	ND	11.49 (10.45–16.26)	9.21 (7.74–10.41)	5.69 (1.09–7.72)	
OR, CIN2+	1.0	0.75 (0.45–1.25)	3.20 (1.72–5.94)	9.08 (3.80–21.69)	<0.001
*p*		0.270	<0.001	<0.001	
Log(all-hr)	Negative *n* = 85	Low *n* = 130	Moderate *n* = 130	High *n* = 131	
Median (range)	ND	3.23 (0.99–4.09)	4.75 (4.10–5.29)	6.02 (5.31–8.53)	
OR, CIN2+	1.0	3.24 (1.83–5.73)	10.08 (5.25–19.35)	5.03 (2.79–9.07)	<0.001
*p*		<0.001	<0.001	<0.001	
Log(HPV16/18)	Negative *n* = 283	Low *n* = 64	Moderate *n* = 64	High *n* = 65	
Median (range)	ND	3.22 (1.17–4.05)	4.76 (4.06–5.25)	5.93 (5.26–8.21)	
OR, CIN2+	1.0	2.16 (1.18–3.94)	7.54 (3.15–18.04)	4.29 (2.10–8.76)	<0.001
*p*		0.012	<0.001	<0.001	
Log(other-hrHPV)	Negative *n* = 219	Low *n* = 85	Moderate *n* = 85	High *n* = 87	
Median (range)	ND	2.92 (0.99–3.70)	4.39 (3.73–5.03)	5.95 (5.06–8.53)	
OR, CIN2+	1.0	0.90 (0.53–1.51)	1.76 (0.99–3.13)	1.14 (0.67–1.94)	<0.001
*p*		0.685	0.053	0.621	

^1^ ORs and 95%CI were calculated using logistic regression. ^2^ *p*-trend from a one-degree-of-freedom trend test.

**Table 4 pathogens-12-00063-t004:** Performance of cytology, hrHPV test, PAX1^m^, HPV viral load of all hrHPV and HPV16/18 for detecting CIN2+.

Tests	Se% (95%CI)	Sp% (95%CI)	PPV %(95%CI)	NPV %(95%CI)	YI (95%CI)
TCT ≥ ASCUS	54.5 (48.9–60.1)	72.6 (64.9–79.4)	80.2 (75.5–84.2)	44.0 (40.3–47.8)	0.272 (0.173–0.359)
*p*	0.029	0.001	0.012	0.899	0.291
* PAX1^m^ ≤ 10.63	46.4 (40.8–52.0)	86.6 (80.3–91.5)	87.6 (82.3–91.4)	44.3 (41.4–47.3)	0.330 (0.245–0.400)
hrHPV(+)	90.6 (86.8–93.6)	35.0 (27.6–43.0)	73.9 (71.5–76.2)	64.7 (55.1–73.3)	0.256 (0.177–0.339)
*p*	<0.001	<0.001	<0.001	<0.001	0.302
HPV16/18(+)	50.2 (44.5–55.8)	79.0 (71.8–85.1)	82.9 (77.8–87.0)	43.8 (40.5–47.2)	0.291 (0.193–0.371)
*p*	0.349	0.058	0.104	0.816	0.458
Log(all-hr) > 3.32	78.1 (73.1–82.5)	56.7 (48.5–64.6)	78.5 (75.2–81.6)	56.0 (49.8–62.0)	0.347 (0.252–0.440)
*p*	<0.001	<0.001	<0.001	<0.001	0.782
Log(16/18) > 2.34	48.0 (42.4–53.6)	84.1 (77.4–89.4)	86.0 (80.7–89.2)	44.3 (41.2–47.4)	0.320 (0.231–0.396)
*p*	0.729	0.557	0.548	0.998	0.840

* All indicators were compared with PAX1^m^.

## Data Availability

The datasets generated and/or analyzed during the current study are available from the corresponding author upon reasonable request.

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
