# Peer review of "Association and Effectiveness of PAX1 Methylation and HPV Viral Load for the Detection of Cervical High-Grade Squamous Intraepithelial Lesion"

_pathogens, 2022, doi:10.3390/pathogens12010063_

Round 1

Reviewer 1 Report

This is an interesting comparison of methylation versus HPV viral load in cervical dysplasia.

1. Title - indicate that it is cervical dysplasia

2. What about patients without abnormal screening? How did these tests perform?

3. A lot of CIN2 versus CIN3 comparisons. Not very relevant and may not be accurate.

4. Figure 1 . I do not see a difference between CI1 and CIN2 for PAX. But there is for HPV VL. Is this not important?

5. Table 3. Lots of information. Probably too much.

6. Table 4 is complex. PAX and VL seem to have very similar test parameters.

5. What are the costs of the 2 tests that are compared here?

Author Response

please find the attached reply

Reviewer 2 Report

There have been several similar articles related to PAX1 methylation in HPV infected CIN lesions. 

The relationship between HPV viral load and CIN grading and progression is well understood,so it is not necessary to cite more than 10 articles to illustrate it.

Table 1 contain too much content,it should be divided to several tables of figures to make it clear.

Table 3 and Table 4 have same problem,need to be revised.

There are some writting problems need to be improved, such as “HPV 16/18 women“ should be cervical HPV16/18 positive women.

Author Response

please find the attached reply
